# Orbital Angular Momentum Mode Sensing Technology Based on Intensity Interrogation

**DOI:** 10.3390/s22051810

**Published:** 2022-02-25

**Authors:** Churou Huang, Guoxuan Zhu, Zhiyong Bai, Jiayan Chen, Zheng Huang, Rui Liu, Luping Wu, Shen Liu, Cailing Fu, Yiping Wang

**Affiliations:** 1Key Laboratory of Optoelectronic Devices and Systems of Ministry of Education/GuangDong Province, College of Physics and Optoelectronic Engineering, Shenzhen University, Shenzhen 518060, China; 1900453003@email.szu.edu.cn (C.H.); zhugxuan@szu.edu.cn (G.Z.); 1900453055@email.szu.edu.cn (J.C.); 1910454042@email.szu.edu.cn (Z.H.); 2060453053@email.szu.edu.cn (R.L.); 2070456106@email.szu.edu.cn (L.W.); shenliu@szu.edu.cn (S.L.); fucailing@szu.edu.cn (C.F.); ypwang@szu.edu.cn (Y.W.); 2Shenzhen Key Laboratory of Photonic Devices and Sensing Systems for Internet of Things, Guangdong and Hong Kong Joint Research Centre for Optical Fibre Sensors, Shenzhen University, Shenzhen 518060, China

**Keywords:** orbital angular momentum, chiral long-period fiber grating, temperature sensor, intensity interrogation

## Abstract

A novel optical fiber sensing technology based on intensity distribution change in orbital angular momentum (OAM) mode is proposed and implemented herein. The technology utilizes a chiral long-period fiber grating (CLPFG) to directly excite the 1st-order OAM (OAM_1_) mode. The intensity changes in the coherent superposition state between the fundamental mode and the OAM_1_ mode at the non-resonant wavelength of the CLPFG is tracked in order to sense the external parameters applied to the grating area. Applying this technology to temperature measurement, the intensity distribution change has a good linear relationship with respect to temperature in the range of 30 °C to 100 °C. When the intensity was denoted by the number of pixels with a gray value of one after binarization of collected images, the sensitivity was 103 px/°C and the corresponding resolution was 0.0097 °C. Meanwhile, theoretical and experimental results show that the sensitivity and resolution can be further improved via changing the area of the collected image. Compared with sensing methods based on spiral interference pattern rotation in previous work, this sensing technology has the advantage of exquisite structure, easy realization, and good stability, thus making it a potential application in practices.

## 1. Introduction

Orbital angular momentum (OAM) beam [1] is a kind of beam with a helical phase front, which is generally expressed by exp(ilφ), where l represents the topological charge number and φ is the azimuth angle. This special phase structure induces a phase singularity, and a donut-shaped intensity profile of an OAM beam. Due to the unique structure, the OAM beam has been widely utilized in optical communications [2,3], optical manipulation [4], optical imaging [5], and especially in optical sensing. For instance, the OAM beams have been used to measure the rotation of an object based on Doppler shift without the need for complicated object image reconstruction [6], and the OAM-based technology [7] expands a new optical degree of freedom perspective to detect the lateral motion in an arbitrary direction. Also, magnetic field measurement [8] applies OAM modes with opposite propagating directions that interfere to form petal-shaped images. It turns out that the rotation angle of the petal-shaped images changes with the change in the magnetic field and the sensitivity is 28°/T. The method is suitable for high-precision measurement applications, however, a higher cost and a larger volume spatial OAM generator was used in the experiment. Recently, optical fiber sensors based on OAM modes [9] have been proposed and achieved by tracking the rotation angle of the spiral interference patterns resulting from the co-axis interference between OAM modes and Gaussian beams. The sensitivity of temperature is 87.09°/°C. The advantage of the OAM-based fiber sensor is mainly to measure the changes in the external parameters through the rotation angle of spiral interference fringes, which avoids the reading error of non-integer fringes in the traditional interference measurement. The sensitivity and resolution are generally high. At present, optical fiber sensors based on OAM mainly use the structure of double optical path interference. The OAM sensing technology based on the interference of the Gaussian beam and the OAM beam has been reported in literature [10,11,12]. The phase difference of the two optical paths is changed by heating the sensing fiber, resulting in the rotation of the spiral interference pattern. The temperature change is tracked by monitoring the rotation angle of the spiral interference pattern. The difference is that their OAM generators and demodulation methods are different. The OAM generators are a hologram [10], a spiral phase plate [11], and long-period fiber grating and a rotator [12] respectively. The disadvantages of this structure are that the interference pattern is easily affected by external parameters, is unstable, and is difficult to collect, which has an adverse impact on its practical application.

In this paper, a single optical path sensing technique is proposed by tracking the intensity change in the superimposed state between the fundamental mode (OAM_0_) and the 1st-order OAM (OAM_1_) mode. The single optical path was composed by a chiral long-period fiber grating (CLPFG) and a section of two-mode fiber (TMF) [10]. The CLPFG [11,12,13,14,15,16] was used to directly excite the OAM_1_ mode. At the non-resonant wavelength of CLPFG, the OAM_0_, and OAM_1_ mode coexisted and then were transmitted in TMF. At the output end of TMF, the coherent superimposition of the two modes was obtained. When the environmental variables were applied to the CLPFG, the energy ratio between the OAM_0_ and OAM_1_ mode changed and then influenced the intensity of the superimposed mode. The sensing technology was applied in the temperature measurement and the consequence shows that the sensitivity and the resolution are 103 px/°C and 0.0097 °C in the range of 30 °C to 100 °C, respectively. Meanwhile, the linear fitting results show that the intensity change has a linear relationship with temperature applied. By changing the image area, the sensitivity can be adjusted from 1 px/°C to 103 px/°C.

## 2. Principle

The schematic diagram of the operation principle is shown in Figure 1. The sensing system is composed of an OAM mode generator and a section of TMF. In this work, the OAM mode generator is a CLFPG, which was fabricated by twisting molten single-mode fiber (SMF-28e) at high temperatures. As illustrated in the inset of Figure 1, when the grating pitch of the CLPFG is 530 µm, the resonant wavelength is 1572.5 nm and the dip loss is 31.2 dB, thus directly coupling the OAM_0_ to the OAM_1_ mode with high efficiency. When the incident light is located at the resonant wavelength of CLPFG, the fundamental mode of the fiber, that is, the OAM_0_ mode, is completely transformed into the OAM_1_ mode by the CLFPG. As shown in Figure 2, it can be calculated, by finite element analysis, that TMF (14 µm/125 µm for core/cladding diameter) can support the vector modes of HE_21_^even^, HM_21_^odd^, TE_01_, and TM_01_, which can be superimposed into the OAM_1_ mode. Therefore, the TMF is fused at the back end of CLFPG, which can transmit the OAM mode excited in CLPFG within a certain distance. At the back end of the TMF, only the ring-shape mode profile belonging to the OAM_1_ mode can be seen. When the wavelength of the input light is far from the grating loss dip, the OAM_0_ mode can be seen at the back end of TMF. When the input light has a wavelength within the loss dip bandwidth, a part of the OAM_0_ mode energy is coupled to the OAM_1_ mode. At this time, the OAM_0_ and OAM_1_ mode coexist. After they are transmitted through the TMF, mode interference occurs at the output end, and the coherent superposition state of the two modes is observed. 

Generally speaking, by changing the external environment parameters of the TMF, the phase difference between the OAM_0_ and OAM_1_ mode can be changed, the rotation change in the superposition state can be observed at the output end, and then the sensing measurement can be carried out by tracking the rotation angle. However, the OAM mode generator is an all-fiber device. It is difficult to achieve perfect coaxially distribution in the intensity of the OAM mode. Therefore, when the sensor is performed by tracking the phase shift, except the rotation of the spiral interference pattern, the intensity distribution is changed as well, which brings potential difficulties for accurate image acquisition and accurate measurement. Although the sensing sensitivity is high, it is difficult to achieve the stable acquisition and processing of rotating images, which limits its practical application. Thus, the measurement can be realized by tracking the intensity change in the interference patterns, where the spiral interference patterns only changes in intensity without rotation. Taking temperature as an example, when the external temperature changes, the grating refractive index changes due to the thermal effect and the thermal expansion effect, resulting in the shift of resonant wavelength. Thus, coherent superposition state between the OAM_0_ and OAM_1_ mode is observed in CLPFG at the same wavelength near the resonant wavelength. Different temperatures affect the change in energy ratio. In this paper, by applying variable environmental parameters to the OAM mode generator, that is the CLPFG, the change in the energy proportion of the OAM_0_ and OAM_1_ mode can be induced, due to the response characteristics of the CLPFG. In this work, the measurement of environmental parameters can be realized via tracking the change in the intensity profile of the superimposed modes. 

The inset in Figure 1 shows the transmission spectrum of CLPFG, from which the OAM_0_ and OAM_1_ mode coexist at the non-resonant wavelength λ_1_. The temperature measurement is taken as an example to illustrate the sensing and interrogating technology. When the temperature changes, the energy ratio of the OAM_0_ and OAM_1_ mode changes, resulting in a change in the intensity distribution of the superposition state. As is shown in Figure 3, the numerical calculation of sensing principle can be obtained. First, the OAM_1_ mode measured at the resonant wavelength and the OAM_0_ mode measured without writing CLPFG are regarded as the calculation basis. Second, by changing the energy ratio of the two modes, the intensity changes in the superposition state can be calculated, as shown in Figure 3a. In the calculation process, the superimposed mode field is binarized, and the total number of pixels with a gray value of one is used to represent the intensity. Then, the change in intensity, with respect to energy ratio, is fitted and plotted in Figure 3b, and the sensitivity and linearity (R^2^) are predicted. It is worth noting that the size of the acquired image and the setting of the threshold in the binarization process will affect the results. As shown in Figure 3c, S1–S6 represent different imaging areas, respectively, as follows: 0.41 mm^2^, 1.21 mm^2^, 3.27 mm^2^, 6.88 mm^2^, 11.58 mm^2^, and 18.69 mm^2^. When the threshold is greater than 10 (grayscale value 0–255), the sensitivity and linearity of the intensity-to-energy ratio response tend to be stable under different imaging areas, indicating that an appropriate threshold can eliminate the influence of background noise. At this time, the corresponding threshold can be determined. For different imaging areas, the sensitivity is different. Figure 3d redraws the sensitivity of the different imaging areas at a threshold of 17. The sensitivity gradually increases as the imaging area increases.

## 3. Sensing Characteristics

The experimental setup of the proposed sensing method is shown in Figure 4, which is mainly composed of a tunable laser emitting single-wavelength light, a CLPFG that excites the OAM_1_ mode, and a CCD that collects the intensity distribution. The tunable laser (Agilent 81940A, USA) has relatively stable power while emitting. When the output laser exceeded 1 hour, the power fluctuation range was ± 0.0075 dB, which can be negligible. The CLFPG, with a resonant wavelength of 1572.5 nm, was fabricated by twisting the hot-melt SMF. As such, a section of TMF was fused at the end of the grating at an interval of 5 mm to transmit high-order modes in order to facilitate the measurement of the mode field distribution and its changes. We used a compact precision temperature control box (LCO 102 SINGLE, ECOM) as the temperature generator. The device can produce temperature changes in the range of 0–100 °C, with an accuracy of 0.01 °C. The CLPFG of the measuring section was placed in a closed temperature control box, and both ends of the optical fiber passing through the temperature control box were fixed on the optical fiber clamp. Through heat conduction, CLPFG can obtain the same temperature change as the temperature control box, so as to study the temperature sensing performance of the sensor. The operation wavelength was selected to be 1570.5 nm, in which the OAM_0_ and OAM_1_ mode were in the state of coexistence. It cannot be ignored that all-optical fiber devices are fixed on the platform, which helps to collect the stable images. The temperature rose from 30 °C to 100 °C, at an interval of 5 °C, and each temperature remained for 5 min in order to ensure that the grating was in a state of constant temperature. The CCD (Hamamatsu, C12741-03, Japan) has a resolution of 640 × 512 px and the size of the single-pixel is 20 µm × 20 µm. It recorded the changes in the superimposed mode intensity. According to the display, the intensity of the images must not be exposed, so as to achieve a good match between the intensity distribution and the gray value of images. What’s more, the type of images collected by the camera is uint8, which has a gray-level range of 0–255.

The detailed experiment of temperature measurement was performed as follows. First, the intensity distribution of the coherent superposition modes, with an imaging area of 7.83 mm^2^, were collected when the temperature increased from 30 °C to 100 °C. Second, the collected images were binarized under the threshold with a gray value of 41. The collected images and their corresponding binarized results are shown in Figure 5a. Third, the number of pixels with a gray value of one, representing intensity information, was counted and is plotted here in Figure 5b. And then, we repeated the measurement step for the cooling process, and plotted the intensity distribution pattern and experiment data in Figure 5a,b respectively. In Figure 5a, the intensity distribution pattern of the interference pattern does not rotate, and only shows the change in intensity. In Figure 5b, the intensity change data measured during the heating and cooling show linear monotonicity. Through fitting, the temperature sensitivity and linearity under the test conditions are 48 px/°C and 0.995, respectively.

In the process of temperature detection, different operating wavelengths and different binarization thresholds affect the sensitivity of the test. Therefore, the effects of these two factors on the sensitivity were studied. We repeated the temperature measurement processing and scanned the operation wavelength from 1560 nm to 1580 nm, with an interval of 0.5 nm. The recorded intensity graphs of the superimposed modes were binarized with the threshold from 20 to 50. The variation trend of the measured sensitivity with wavelength and threshold is shown in Figure 5c. The temperature response sensitivity generally shows a trend of first increasing and then decreasing, for which, as the operation wavelength is close to the resonant wavelength, the slope of the energy ratio first increases and then decreases. In the meantime, due to the defects in the collected images, the linearity of the different wavelengths under the different thresholds fluctuates. Therefore, choosing an appropriate threshold can improve the stability and accuracy of the measurement. Generally speaking, when the threshold is 41, the temperature sensitivity at the wavelength of 1571 nm is higher and the linearity is also improved, respectively, about 48 px/°C and 0.995.

In theory, the sensitivity can be enhanced by changing the number of participating pixels. Thus, we studied the evolution of sensitivity under different imaging areas. The intensity distributions shown in Figure 6a, represent the different imaging areas of 0.15 mm^2^, 0.85 mm^2^, 2.31 mm^2^, 3.76 mm^2^, 7.83 mm^2^, 9.26 mm^2^, and 16.38 mm^2^, respectively. In Figure 6b, the corresponding sensitivities are 1 px/°C, 5 px/°C, 14 px/°C, 27 px/°C, 48 px/°C, 53 px/°C, and 103 px/°C, respectively. When the imaging area was expanded to 16.38 mm^2^ within the limited experiment condition, we repeated the experiment setup above. The corresponding results are shown in Figure 6c. The temperature sensitivity 103 px/°C was achieved at a threshold of 60, which is larger than the sensitivity of the imaging area of 7.83 mm^2^. Since the CCD can recognize one pixel at least, the theoretical temperature resolution of the sensing technology is 0.0097 °C. In addition, according to the method obtained in Figure 5c, we explored the evolution process of the different working wavelengths under the different thresholds in the imaging area of 16.38 mm^2^, as shown in Figure 6d. By optimizing the optical aperture of the imaging element, the highest sensitivity (of about 809 px/°C) may be obtained by further enlarging the imaging area. In the experiment, the sensitivity can be improved from 1 px/°C to 103 px/°C, under the current system conditions.

At present, the representatives of fiber-based temperature measurement technology include wavelength demodulation technology and intensity demodulation technology. The general wavelength demodulation technology has a sensitivity of 132.8 pm/°C [17] and a resolution of about 0.1 °C. The temperature demodulation technology based on intensity [18] has low sensing resolution, i.e., 0.5 °C. However, the theoretical resolution of the temperature sensor based on OAM [9] can reach 10^-7^ °C, and the actual measurement can reach 10^-5^ °C. The temperature sensing technology based on intensity demodulation of OAM interference that is proposed in this work has better stability and is more easily applied, although its sensitivity is lower than that of the phase-shift OAM-based sensor.

## 4. Conclusions

An optical fiber sensing technology based on intensity distribution change in OAM mode have been investigated and verified. The results display that the response of energy change to temperature is linear, with a sensitivity of 103 px/°C and a resolution of 0.0097 °C. Different operation wavelengths have different sensitivities, which provides a method to adjust the sensitivity. Moreover, the larger the imaging area, the higher the sensitivity and resolution of the sensing system. By using a CCD with higher resolution and increasing the imaging area, the sensitivity and resolution can be improved. This method has a simple structure, small size, high stability, and it has better application potential. In practical application, because CLPFG has a response to a variety of parameters, it brings the problem of cross sensitivity, which can be solved by studying the multi-parameter simultaneous measurement technology based on mode profile change.

## Figures and Tables

**Figure 1 sensors-22-01810-f001:**
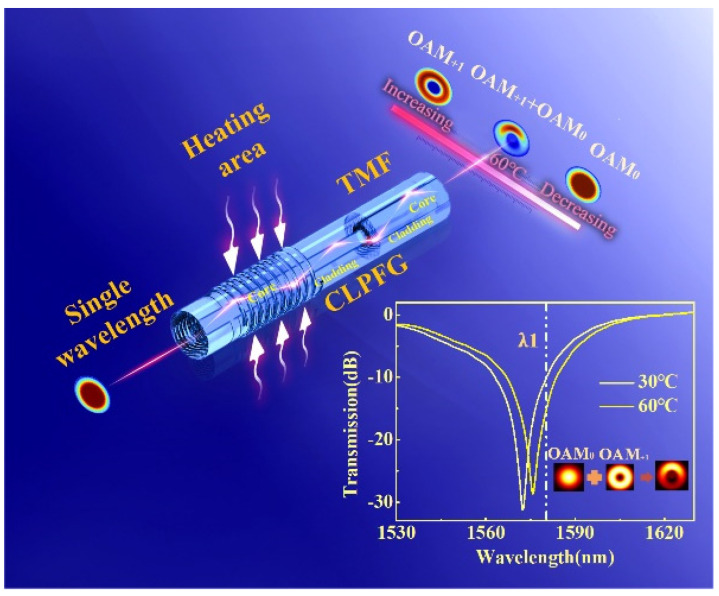
Schematic diagram of sensing principle. TMF: two−mode fiber; the inset is the transmission spectrum of CLPFG at a different temperature.

**Figure 2 sensors-22-01810-f002:**
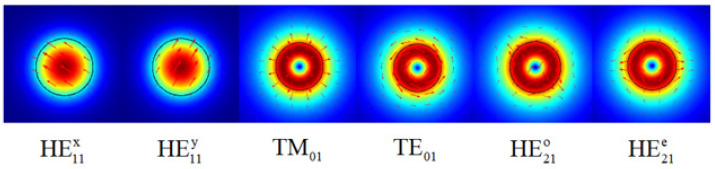
Schematic diagram of TMF transmission mode obtained by simulation.

**Figure 3 sensors-22-01810-f003:**
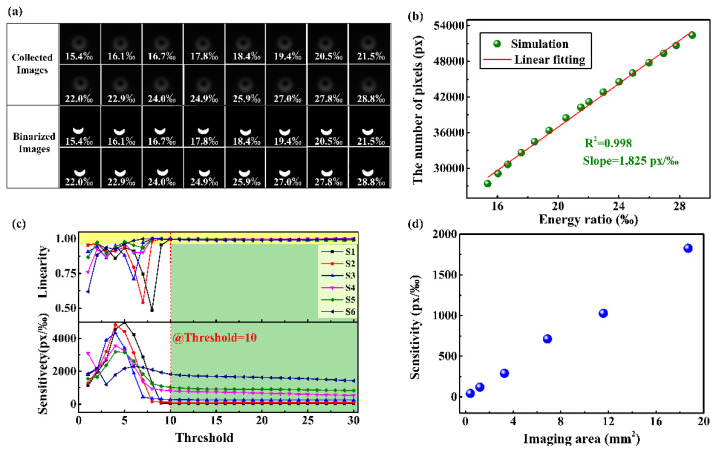
Numerical calculation of sensing principle. (**a**) The collected and binarized images of the intensity distributions in different energy ratio between the OAM_0_ and OAM_1_ mode; (**b**) the relationship between the energy ratio and the number of pixels with the gray value of 1; (**c**) the evolution of sensitivity and linearity of the sensor with the changes in imaging area and thresholds, and the size of imaging areas denoted by S1–S6 are 0.41 mm^2^, 1.21 mm^2^, 3.27 mm^2^, 6.88 mm^2^, 11.58 mm^2^, and 18.69 mm^2^, respectively; (**d**) the relationship between sensitivity and image area.

**Figure 4 sensors-22-01810-f004:**
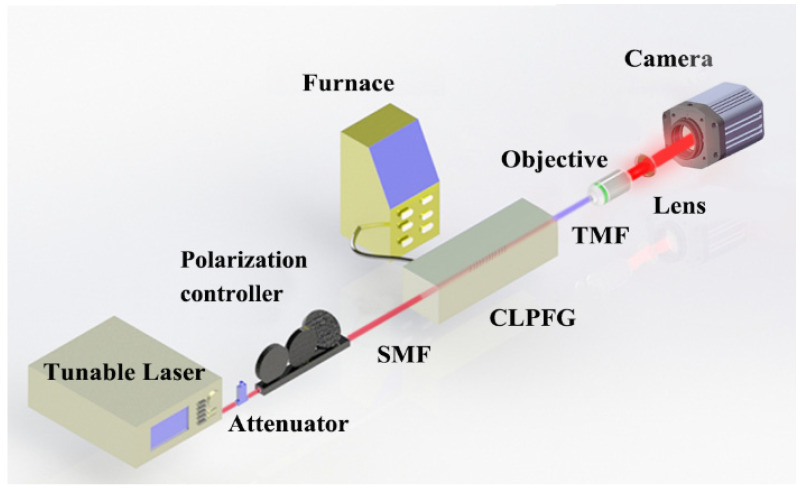
The experiment system of temperature measurement.

**Figure 5 sensors-22-01810-f005:**
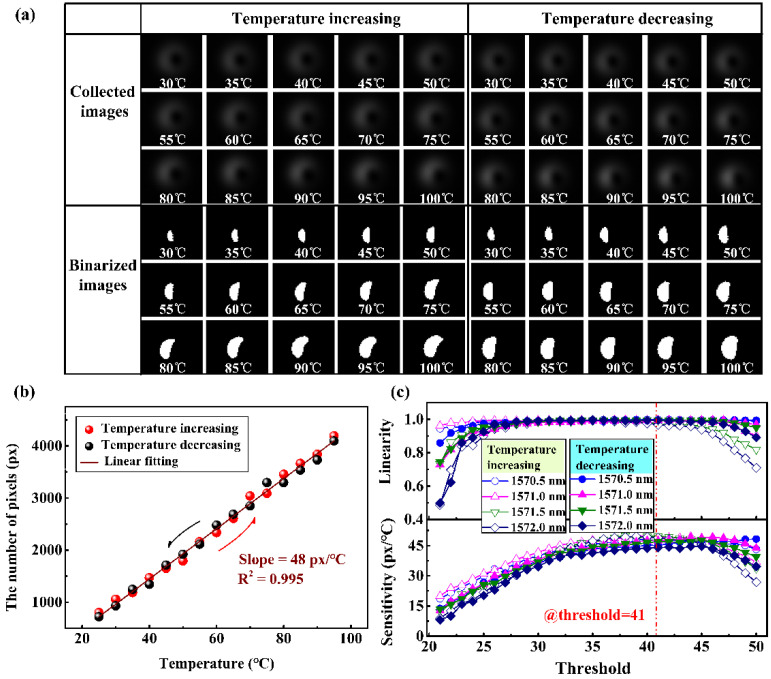
Experimental measurement results. (**a**–**c**) Experiment data with imaging area 7.83 mm^2^ with temperature increasing and decreasing: (**a**) the collected and binarized images of the intensity distributions from 30 °C to 100 °C at 1571 nm; (**b**) experiment data of the temperature test and its linear fitting; (**c**) the evolution of sensitivity and linearity with different operation wavelength and threshold.

**Figure 6 sensors-22-01810-f006:**
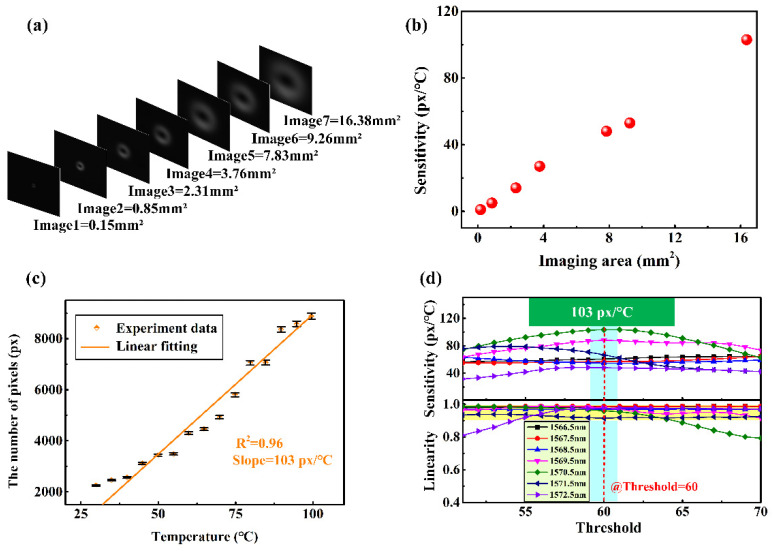
(**a**) The collected images with different imaging area; (**b**) the sensitivity changes with respect to the different imaging area. (**c**,**d**) Experiment data with imaging area 16.38 mm^2^ in temperature increasing: (**c**) experiment data of temperature test and its linear fitting; (**d**) the evolution of sensitivity and linearity with different operation wavelength and threshold.

## Data Availability

Data underlying the results presented in this paper are not publicly available at this time but may be obtained from the authors upon reasonable request.

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
