# Peer review of "Orbital Angular Momentum Mode Sensing Technology Based on Intensity Interrogation"

_sensors, 2022, doi:10.3390/s22051810_

Round 1
Reviewer 1 Report
The authors report on a fiber optical temperature sensor that is based on the interference of two modes with and without an orbital angular momentum. The interference pattern is recorded by a CCD camera, and the temperature is finally computed from the image data. The experimental results sound reasonable, and show that temperature measurement is indeed possible by this method.
However, I have some doubts that this is an efficient method of temperature measurement, as it requires to record images with several hundred kilo pixels, and perform image processing in order to obtain a single scalar value (temperature). One possible argument for measurement techniques utilizing interference is their high precision. But the data presented in this manuscript do not show such a high precision. Moreover, the interference patterns presented in Fig. 4 do not show a rotation with increasing temperature as one would expect as an effect of phase shifts. Instead, as explained in the text, the effect is based on a variation of the intensity ratios of the modes which is not the typical (and efficient) way to operate interference based sensors.
Moreover, the information on the system, and the explanation of its operation are rather superficial. There are no detailed data of the TMF, and CLPFG, like (core) diameter, refractive index profile, refractive index modulation and period of the grating, effective indices of the modes, etc. There is no information on the light power or intensity, the sensitivity of the camera, and how they achieved a good fit of the intensity distribution of the image and the 256 gray levels of the sensor.
Finally, the temperature resolution of 0.0097 °C sounds good, but there is no information on the actual precision and reproducibility of the temperature measurement, which is important to evaluate a sensor. There are a lot of fiber optical temperature sensors presented in the literature. Is there a special case of application where the sensor shown here has advantages? There is no hint on that in the manuscript.
In conclusion, I propose to reject the manuscript.
Reviewer 2 Report
The manuscript sensors-1557215 has been submitted as a communication. The report mainly presents a particular optical fiber sensing technique to describe external parameters with an influence in the intensity distribution change of orbital angular momentum mode in propagation through the system. Please see below a list of comments for the authors: 1. The authors wrote “The sensing technology was applied in the temperature measurement and the consequence shows that the sensitivity and the resolution are 103 px/°C and 0.0097 °C in the range of 30 °C to 100 °C, 67 respectively.” However, these parameters correspond to the particular system of fibers and stability in the laser irradiation; which is not described in the work. 2. Error bar and statistical measurements are mandatory to be reported. 3. It is suggested to split the collective citations with individual expressions to justify each reference selected for this communication. 4. The temperature emitted by the furnace is different from the temperature in the fiber. How is measured the temperature in the fiber to validate the calibration of the system proposed? Please describe. 5. How is the influence of the incident polarization and the potential evolution of the gaussian beam profile during the propagation in the system in the main observations? 6. In order to see the value of the main findings, the advantages and disadvantages of this technique based on OAM modulation must be confronted with updated publications related to high sensitivity sensing by different technologies. You can consider for instance the assistance of chaos theory for temperature sensing doi:10.3390/pr8111377 or fiber-optic microsensing https://doi.org/10.1007/s13320-021-0632-7 7. How is determined the threshold in the binarized process of the images? 8. How is controlled the reproducibility in the OAM mode generation? 9. The expression “the change of the energy proportion of the OAM0 and OAM1 mode can be induced due to the response characteristics of the CLPFG” must be explained with better details. 10. How is identified that the change in the beam profile is exclusively due to a change in temperature and not to other parameters? Please argue considering real applications.Author Response
Please see the attachment.

Round 2
Reviewer 1 Report
The authors have provided improvements according to my previous comments.
Reviewer 2 Report
First at all, I must say that it can be clearly noticed the value of this brilliant work, and I appreciate the effort of the authors to clarify the issues raised in the review stage. However, some fundamental points have not been addressed within the manuscript and only a partial response is in the reply to reviewers. In my opinion, the presentation of the text could be still improved according to previous recommendations with proper discussion and/or citation if needed. Please see below the list that currently remains in this second round of review:
*4. The temperature emitted by the furnace is different from the temperature in the fiber. How is measured the temperature in the fiber to validate the calibration of the system proposed? Please describe.
*6. In order to see the value of the main findings, the advantages and disadvantages of this technique based on OAM modulation must be confronted with updated publications related to high sensitivity sensing by different technologies. You can consider for instance the assistance of chaos theory for temperature sensing doi:10.3390/pr8111377 or fiber-optic microsensing https://doi.org/10.1007/s13320-021-0632-7
*8. How is controlled the reproducibility in the OAM mode generation?
*10. How is identified that the change in the beam profile is exclusively due to a change in temperature and not to other parameters? Please argue considering real applications.
Round 3
Reviewer 2 Report
The authors have clarified more about their research. However, in particular for the points 6 and 10 raised in the first round of the review stage, the authors have given partial explanations without a citation in the reply to reviewers with no substantial changes in the manuscript.
In my opinion, the real sensing applications of the proposed methodology for a given system and proper critical discussion of the analytical performance correspond to aspects that are crucial for the journal. Then, the confrontation of specific parameters related to sensing performance and sensibility of other updated techniques has been requested for addressing point 6.
Moreover, my main concern related to the beam profile exclusively associated with temperature could be initially solved if data plotted in figure 5 are acquired with increments and decrements of temperature and not only in a monotonic trace.
Round 4
Reviewer 2 Report
The authors have clarified within the text their main findings and then in my opinion, this work can be considered for publication in present form.